# PLaID++: A Preference Aligned Language Model for Targeted Inorganic Materials Design

Andy Xu [1]   Rohan Desai [1]   Larry Wang [1]   Ethan Ritz [1]   Gabriel Hope [2]

## Abstract

Reinforcement Learning from Verifiable Rewards (RLVR) has emerged as a promising approach to improve correctness in LLMs, however, in many scientific problems, the objective is not necessarily to produce *the* correct answer, but instead to produce a diverse array of candidates which satisfy a set of constraints. We study this challenge in the context of materials generation. To this end, we introduce PLaID++, an LLM post-trained for stable and property-guided crystal generation. We find that applying naive preference optimization to a coordinate-based crystal representation leads to mode collapse. Hence, we introduce a compact, symmetry-informed Wyckoff text representation which improves computational efficiency and encourages generalization from physical priors. By encoding symmetry constraints directly into text and guiding model outputs towards desirable chemical space, PLaID++ generates structures that are thermodynamically stable, unique, and novel at a $>50\%$ greater rate than prior methods. We further demonstrate that unified training across conditional and unconditional tasks are mutually beneficial in data-sparse regimes. Our work demonstrates the potential of adapting post-training techniques from natural language processing to materials design, paving the way for targeted and efficient discovery of novel materials.

## 1. Introduction

The discovery of new solid-state materials is the foundation of many transformative technologies including solar cells (Green et al., 2014), batteries (Zhao et al., 2020), and car-bon capture (Sriram et al., 2024a). However, the search for new materials is constrained by the immense scale of chemical space—previous explorations have only uncovered a fraction of the total number of potential stable inorganic compounds (Davies et al., 2016). Generative models offer a promising avenue for accelerating technological breakthroughs by efficiently discovering novel and stable structures in unexplored regions of chemical space.

Previous work has applied variational autoencoders (Xie et al., 2021) and denoising models (Zeni et al., 2025; Jiao et al., 2023; Miller et al., 2024) to generate stable and novel structures. However, these works either do not explicitly encode crystallographic symmetry or use computationally inefficient and complex representations, limiting the efficacy of their models.

Meanwhile, language models (Gruver et al., 2024; Sriram et al., 2024b) have emerged as a promising alternative. Their pretrained knowledge makes them data-efficient, often requiring far fewer domain-specific examples than training models from scratch (Gruver et al., 2024). Moreover, their natural-language interface allows one model to be applied to many tasks, including unconditional and conditional generation, infilling, and crystal structure prediction (Gruver et al., 2024).

Symmetry is a defining aspect of crystal structures. The set of rotations, reflections, inversions, and translations exhibited by a crystal lattice form its *space group*. A crystal's space group and symmetries are not merely a mathematical construct but critical to many optical, electrical, and magnetic properties like piezoelectricity (Malgrange et al., 2014; Yang et al., 2005). One way to define a crystal's structure in terms of its symmetries is via Wyckoff positions, whereby one can specify a few key atomic coordinates and have the remaining atomic positions defined through the application of symmetry operations (Hahn, 1983). We train an LLM to learn and exploit structural parameters from the symmetries in each crystal. By allowing the model to discover patterns implicitly through training on Wyckoff-based text representations, we see significant improvements in generation performance.

To guide our search space towards chemically useful struc-

[1]Harvey Mudd College, Claremont, CA, USA [2]Swarthmore College, Swarthmore, PA, USA. Correspondence to: Andy Xu <andxu@hmc.edu>.

*Proceedings of the $43^{rd}$ International Conference on Machine Learning*, Seoul, South Korea. PMLR 306, 2026. Copyright 2026 by the author(s).

tures, we introduce Reinforcement Learning from Interatomic Potentials (RLIP), a reinforcement learning framework for physically grounded materials generation. To do so, we adapt Direct Preference Optimization (DPO) (Rafailov et al., 2023), a computationally efficient reinforcement learning method. Our pipeline is as follows: first, we perform supervised finetuning on a base LLM with Wyckoff-based text encodings of crystals. Then we further fine-tune the LLM via multiple rounds of DPO on generated structures categorized by their stability, novelty, and space group. To prevent entropy collapse and increase the diversity of generated structures, we increase the sampling temperature across successive iterations of DPO.

By incorporating inherent crystal symmetries and the feedback of a Machine Learning Interatomic Potential (MLIP), we significantly increase the rate of stable, unique, and novel (S.U.N.) materials. Our experiments demonstrate that our method **PLaID++ generates materials at a state of the art stability rate and generates S.U.N materials at a ∼50% higher rate** than prior models while retaining the flexibility of natural language prompting. By using representations that encourage learning from physical priors and diversity-aware training, we also highlight the effectiveness of reinforcement learning for materials discovery, **achieving ∼115% and ∼50% improvements in unconditional and space group conditioned generation, respectively, compared to fine-tuning alone.** Our contributions are as follows:

1. We introduce Reinforcement-Learning from Interatomic Potentials (RLIP), a diversity-aware framework for fine-tuning LLMs for material generation.

2. Motivated by mode collapse after preference optimization on a coordinate-based representation, we develop a novel, symmetry-informed text representation for crystal structures which is compact, performant, and physically-motivated.

3. We demonstrate that unified training across conditional and unconditional generation tasks are mutually beneficial for data-sparse regimes. We achieve state-of-the-art results for generating novel and stable materials in both settings.

## 2. Related Work

**Variational Autoencoders (VAEs)** were among the early approaches to crystal structure generation, using learned latent spaces to produce physically plausible atomic arrangements (Xie et al., 2021; Court et al., 2020). Specifically, CDVAE introduced a continuous representation of crystal structures, however, VAEs face challenges in generating structures with strict crystallographic constraints, as their representations often lack direct symmetry considerations.

**Diffusion Models** like DiffCSP and FlowMM iteratively denoise atomic positions to generate crystal structures, achieving higher stability and validity than earlier VAE approaches (Jiao et al., 2023; Miller et al., 2024). While these models excel at capturing complex chemical landscapes, they face significant inference cost due to multi-step denoising, which limits their scalability for rapidly screening a broad sample of material candidates.

**Language Models** are scalable and capable when leveraging extensive pre-training on vast text corpora. In particular, CrystalLLM (Gruver et al., 2024) pioneered the use of fine-tuned LLMs for crystal generation, showcasing the effectiveness of text-based representations for materials prediction.

**Symmetry-aware Language Models** have introduced Wyckoff-position-based representations, leveraging inherent crystallographic symmetries to more effectively constrain the generative process within physical reality. WyFormer and CrystFormer (Kazeev et al., 2025; Cao et al., 2024) utilized transformer-based architectures to predict atomic positions conditioned explicitly on predefined space groups and chemical formulas. More recently, CrystalICL (Wang et al., 2025) pairs a space-group-based crystal tokenization with retrieval-selected few-shot demonstrations to enable in-context learning for conditional generation. Although these symmetry-aware methods showed improvements in structural validity, their reliance on explicit formula and space group constraints considerably limits their utility in exploratory *de novo* generation tasks.

## 3. Background

### 3.1. Crystallography

Crystal structures are periodic arrangements of atoms defined by repeating unit cells. Each unit cell is fully described by lattice parameters—consisting of side lengths $(l_1, l_2, l_3)$ and angles $(\theta_1, \theta_2, \theta_3)$—and atomic positions within the cell, characterized by their fractional coordinates $(x_i, y_i, z_i)$ and elemental identities $e_i$.

The properties of crystals are heavily influenced by their underlying symmetry, captured mathematically by their space groups. Each space group comprises a set of symmetry operations, such as rotations, reflections, and inversions, that map the crystal structure onto itself. Atomic positions consistent with these symmetry operations are defined through Wyckoff positions (Hahn, 1983), significantly reducing the degrees of freedom required to specify a structure. Consequently, employing Wyckoff positions in generative modeling can greatly enhance the validity and physical plausibility

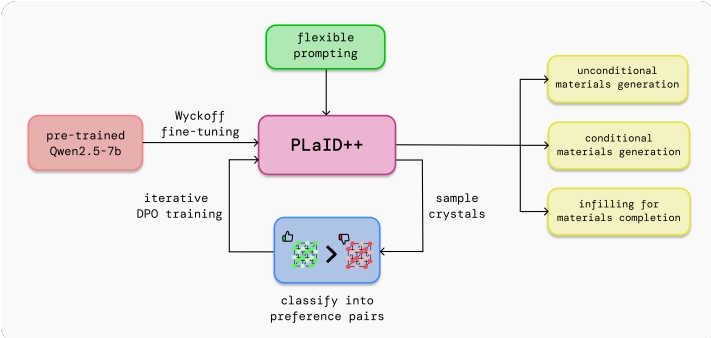

*Figure 1.* Overview of the PLaID++ pipeline, highlighting Wyckoff fine tuning and iterative DPO.

of generated structures. We present additional mathematical background on Wyckoff positions in Appendix 7.8.

In computational materials science, thermodynamic stability is a critical metric which can be described using the energy above hull ($E^{\text{hull}}$), which quantifies how energetically favorable a material is compared to known competing phases (Sun et al., 2016). Materials with $E^{\text{hull}} \leq 0$ eV/atom at a given temperature and pressure are considered thermodynamically stable, while those slightly above zero (typically $E^{\text{hull}} < 0.1$ eV/atom) are metastable and potentially synthesizable (Jain et al., 2013).

### 3.2. Reinforcement Learning for LLM Fine-tuning

RL methods align LLMs to human preferences by maximizing a learned reward while regularizing toward a reference model to prevent mode collapse (Bai et al., 2022). Actor–critic variants such as PPO reduce gradient variance but add overhead via explicit reward and value models (Schulman et al., 2017). Direct Preference Optimization (DPO) bypasses these components by fitting an implicit reward from preference pairs and optimizing a KL-regularized objective directly (Rafailov et al., 2023; Widatalla et al., 2024). DPO uses observed preference pairs of responses $(y_w, y_l)$, generated from $\pi_{ref}$, where $y_w$ denotes the preferred response. DPO optimizes an implicit reward function $r^*(x, y)$ by maximizing the likelihood of preferences under a Bradley-Terry preference model: $p(y_w \succ y_l \mid x) = \exp(r^*(y_w|x))/(\exp(r^*(y_w|x)) + \exp(r^*(y_l|x)))$. The authors show that a KL-regularized objective can be optimized without directly instantiating $r^*(x, y)$, leading to the following objective:

$$\mathcal{L}_{\text{DPO}} = -\mathbb{E}_{(x,y_w,y_l)}\Big[ \log \sigma \Big( \beta \log \frac{\pi_\theta(y_w \mid x)}{\pi_{\text{ref}}(y_w \mid x)} \\ - \beta \log \frac{\pi_\theta(y_l \mid x)}{\pi_{\text{ref}}(y_l \mid x)} \Big) \Big] \quad (1)$$

To effectively generate stable structures, we employ a rein-

forcement learning framework based on DPO. By curating a dataset of stability and novelty-ranked crystal pairs evaluated via MLIP relaxations, our model iteratively learns to favor generation of structures with lower predicted energies in unseen crystal space, thus aligning generative behavior towards chemically stable and experimentally realizable new materials.

## 4. Methodology

### 4.1. Base Model

Our approach begins with a pre-trained large language model (Qwen-2.5 7B) (Yang et al., 2024), which has demonstrated strong capabilities in text generation and reasoning. Following the methodology outlined by (Gruver et al., 2024), we fine-tune the base version of Qwen-2.5 7B to model the crystal generation process via parameter-efficient fine-tuning (Hu et al., 2021). Unlike prior works that rely on explicit space group constraints or predefined templates, our method enables more flexible and generalizable unconditional and conditional structure generation through the use of a natural language prompt.

The training objective is to maximize the likelihood that the model generates the correct token sequence representing stable crystal structures. We provide the prompt template used to fine-tune our model on both objectives in Figure 2. We fine-tune two versions of the base model: one where the crystal structures are represented as text-based 3D coordinates and another based on a new text-based Wyckoff representation.

### 4.2. Direct Preference Optimization (DPO)

To explicitly align our model toward generating crystals with desired properties, we employ an iterative form of Direct Preference Optimization (DPO) (Rafailov et al., 2023), a reinforcement learning technique that directly optimizes model outputs based on ranked preferences as highlighted by Equation 1. We jointly optimize our model on three explicit

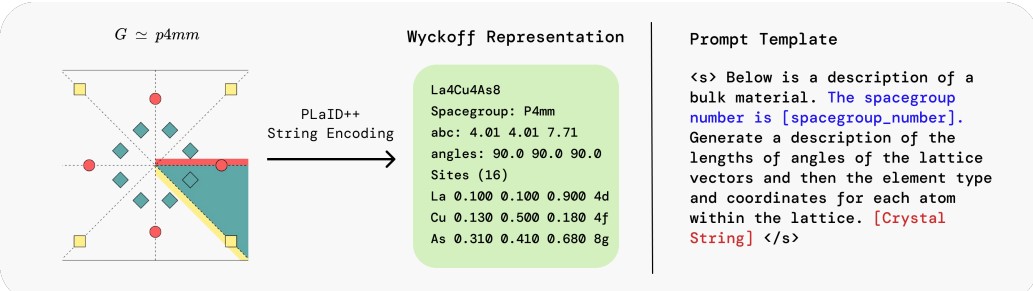

*Figure 2.* **Left:** An example crystal highlighting the $p4mm$ space group symmetry, where colors represent atoms of different elements. We represent the entire asymmetric unit of the crystal in our Wyckoff-based text representation by leveraging the crystal's symmetry. **Right:** The template from which we generate prompts used for training. For conditional generation, we include the blue conditioning information, and for unconditional generation, we remove it from our prompt. In all prompts, the crystal string is replaced with the encoding on the left.

objectives: thermodynamic stability, novelty with respect to a reference crystal set, and space group conditioning.

**Stability** To train the model to prefer stable crystals, we use EquiformerV2 86M (eqV2) (Liao et al., 2024; Barroso-Luque et al., 2024) as a reward signal. eqV2 is a Machine Learning Interatomic Potential (MLIP) that predicts relaxed formation energies for materials, serving as an efficient proxy for stability assessment. To guide the model towards generating more stable materials, we categorize a crystal's energy above the hull into one of three buckets: stable ($\leq$ 0 eV/atom $E^{\text{hull}}$), metastable ($\leq$ 0.08 eV/atom $E^{\text{hull}}$), or unstable ($>$ 0.08 eV/atom $E^{\text{hull}}$). We create stability preference pairs $(x, y_w, y_l)$, where $x$ is the model prompt and $y_w$ and $y_l$ are accepted and rejected crystals. Concretely, we sample preference pairs to form (stable, metastable) and (stable/metastable, unstable) sets, forming a tiered dataset that encodes relative preferences over increasing degrees of thermodynamic stability. This allows us to incorporate fine-grained (stable, metastable) reward signals into our training set without overfitting to exact eqV2-predicted $E^{\text{hull}}$ values and without relying heavily on stable generations from the base model, which are too infrequent to provide sufficient training signal as seen in Appendix 7.2.

**Uniqueness and Novelty** To incorporate novelty into our reward, we construct preference pairs that distinguish between (stable & novel, stable & not novel). Incorporating uniqueness into a pairwise reward is non-trivial and presents many challenges. Namely, since uniqueness is defined at the group level (i.e. whether or not crystals within a set of rollouts are considered the "same" crystal as defined by Pymatgen's StructureMatcher (Ong et al., 2013)), pairwise reward computation of uniqueness is ambiguous. However, we can control sample diversity during rollouts via the sampling temperature. As sampling temperature increases, uniqueness and novelty tend to increase, while stability and validity of generated crystals decrease. Based on this observation, we find that increasing the sampling

temperature across successive iterations of DPO can prevent overoptimization for stability, preventing the model from collapsing on narrow regions of chemical space.

**Space Group** We chose space group as an exemplary conditional generation task since 224/230 space groups have <1,000 samples in our training data. Hence, strong performance improvements in this low data regime is a strong signal for model performance in other data-sparse tasks. To condition our model on generating crystals with a desired space group, we add an additional constraint during preference pair construction. For each space group, we use a conditional generation prompt indicating the target space group number and compare crystals that match this group. Preference pairs are then formed by contrasting (1) stable or metastable crystals with the correct space group against other stable or metastable crystals with the incorrect space group, and (2) stable or metastable structures against unstable structures. This strategy requires stability as a base condition for generation, but also encourages the model to align for structural symmetry.

**Fine-Tuning** We emphasize that both the S.U.N. and space group preference pairs are jointly used in the same DPO objective across the unconditional prompt and space group specific conditional prompts. This joint optimization encourages the model to learn to generate both generally stable crystals and space-group-specific structures in a unified pipeline.

To generate this dataset, we sample 10,000 crystal structures from our fine-tuned Qwen model for unconditional generation and sample 1,000 crystal structures across each of seven space groups. More details about our space group conditional generation is provided in Section 5.1; From here, we categorize our samples into our preference datasets based off the above metrics, resulting in a universal dataset that covers both unconditional and conditional generation.

We apply DPO to fine-tune Qwen on this curated prefer-

ence dataset. We adopt an iterative DPO approach in which $\pi_{\text{ref}} = \pi_{\theta-1}$, with generations from $\pi_{\theta-1}$ used to construct the set of preference pairs for $\pi_\theta$. We have additional information about our DPO hyperparameters, our dataset creation process, as well as other potential DPO variants and ablations in Appendix 7.2.

### 4.3. Wyckoff Representation

When RLIP is naively trained on 3D coordinate representations, PLaID++ models tend to overfit preference pairs and undergo *mode collapse*, producing stable but repetitive crystals (See Section 5.3). We address this failure by introducing a novel Wyckoff representation.

Crystal structures inherently follow symmetry constraints that can be represented by space groups and Wyckoff positions. To improve the model's ability to generate valid and stable structures, our Wyckoff representation encodes these symmetries directly into text. Unlike prior approaches such as DiffCSP++ (Jiao et al., 2024), which rely on searching template structures for Wyckoff sites, our method predicts them directly within the generative process.

Our representation is computationally efficient: on the MP-20 dataset, Wyckoff encodings average 185.5 tokens per crystal compared to 214.7 for coordinate-based strings, a 14% reduction that translates to better sample efficiency and faster training. Moreover, the format scales efficiently as our crystals grow more complex, since additional atoms at the same Wyckoff site require only a multiplicity update rather than new coordinate lines (See Appendix 7.3 for additional details). By tying atoms to Wyckoff sites, a single change propagates through symmetry operations, forcing the model to generalize from physical priors. Together, these properties make our Wyckoff encoding a compact, symmetry-aware foundation that both improves efficiency and mitigates diversity collapse during RL fine-tuning.

## 5. Experiments

### 5.1. Setup

We trained our model on the well-established MP-20 dataset (Xie et al., 2021), a collection of 45,231 inorganic metastable crystalline materials from the Materials Project (Jain et al., 2013) with up to 20 atoms. We follow the methodology of Gruver et al. (2024) by independently fine-tuning a pre-trained Qwen-2.5 7B model using 4-bit quantization and Low-Rank Adapters (LoRA) (Hu et al., 2021), implemented with PyTorch (Paszke et al., 2019) and Transformers (Wolf et al., 2020). Symmetry information for our Wyckoff representation is calculated by Pyxtal (Fredericks et al., 2021). Following supervised fine-tuning, we apply DPO on the generated preference dataset to further guide generation towards stable structures. Full hyperparameters

details are provided in Appendix 7.2.

For unconditional generation, we sample 10,000 structures from each fine-tuned model, parsing a CIF (199) from the generated string. We resample if a CIF cannot be parsed from the string, which guarantees all samples can be interpreted as crystals but does not guarantee validity.

Similarly, for space-group conditioned generation, we sample 1,000 structures for each of seven space groups. These space groups were chosen in accordance with (Zeni et al., 2025) as they represent each of the seven crystal systems with varying levels of symmetry. Specifically, we sample from space groups $P1$, $C2/c$, $Amm2$, $I\bar{4}m2$, $P3$, $P6_3/mmc$, and $F\bar{4}3m$.

To validate the stability and accuracy of our results, we performed energy relaxations on samples generated by our flagship model fine-tuned with DPO and the Wyckoff encoding. Although accurate, Density Functional Theory (DFT) is computationally expensive, scaling typically as $O(N^3)$ with respect to the number of atoms $N$. Therefore, we leverage machine learning interatomic potentials (MLIPs) such as eqV2 (Liao et al., 2024; Barroso-Luque et al., 2024) and eSEN (Fu et al., 2025) as efficient proxies for DFT. These MLIPs predict the relaxed atomic positions and energies with significantly reduced computational overhead. To verify our eSEN results, we compute stability using DFT on a random 1,000 crystal subset of our generated crystals. Detailed information regarding our DFT setup are in Appendix 7.2.

### 5.2. Metrics

To initially evaluate the quality of our generated crystal structures, we focus on structural and compositional validity, as defined by Xie et al. (2021). These metrics provide an effective proxy for assessing the quality of generated crystals before conducting more computationally expensive stability evaluations. We explain more details about these metrics in Appendix 7.1

Our primary metric for evaluating generated crystal structures is the **S.U.N Rate** from Miller et al. (2024), which measures the percentage of crystals that are **stable, unique**, and **novel**. Stability is assessed by comparing a crystal's energy to a convex hull of previously computed energies from Riebesell et al. (2025). Crystals on or below the hull ($\leq 0$ eV/atom $E^{\text{hull}}$) are deemed *stable*. We evaluate **uniqueness**—differentiation from other generated crystals— and **novelty**, which measures diversity from the training data to calculate S.U.N.

For our space group conditional generation task, we adopt the **S.S.U.N Rate** metric from Zeni et al. (2025), which measures the percentage of structures that are the correct symmetry group as deemed by Pyxtal (Fredericks et al.,

*Table 1.* Results for unconditional materials generation on the MP-20 dataset. Our flagship PLaID++ variant uses a Wyckoff text representation, DPO on stability and novelty-based preference pairs, and dynamic temperature adjustment. Column-best figures are in **bold**.

| Method | Params | Validity (%) (↑) | | Stability (%) (↑) | S.U.N. (%) (↑) |
|---|---|---|---|---|---|
| | | Structural | Composition | | |
| CDVAE | – | **100.00** | 86.70 | 1.6 | – |
| DiffCSP | – | **100.00** | 83.25 | 5.0 | 3.3 |
| FlowMM | – | 96.85 | 83.19 | 4.6 | 2.8 |
| FlowLLM | – | 99.94 | 90.84 | 13.9 | 4.7 |
| MatterGen-MP | – | – | – | 13.0 | – |
| CrystalLLM-70B | – | 99.60 | 95.40 | 5.28 | – |
| Jointly-trained ADiT | – | 99.74 | 92.14 | 15.4 | 5.3 |
| | 3D Coord | 98.93 | 88.94 | 7.20 | 2.81 |
| PLaID++ (Non-DPO Types) | Wyckoff | 99.80 | 91.22 | 7.17 | 3.58 |
| | Iterative SFT | 99.85 | 94.54 | 9.85 | 4.13 |
| | 3D Coord | 96.62 | 89.94 | 13.94 | 1.69 |
| PLaID++ (DPO Types) | Wyckoff | 98.81 | 92.57 | 15.59 | 6.25 |
| | Stability + Novelty | 99.78 | 95.64 | 16.72 | 6.22 |
| PLaID++ | – | 99.75 | **97.34** | **22.27** | **7.74** |

2021) and are metastable ($\leq$ 0.1 eV/atom $E^{\text{hull}}$), unique and novel.

Due to compute constraints, we assess stability for unconditional and conditional generation primarily via the eSEN MLIP (Fu et al., 2025). Note that we specifically **use different MLIPs**—eqV2 and eSEN—for preference dataset creation and crystal generation evaluation to avoid reward hacking or overfitting to a specific MLIP.

### 5.3. Results

**Unconditional Generation** We compare our model to seven prior methods, as reported from Joshi et al. (2025): CDVAE (Xie et al., 2021), DiffCSP (Jiao et al., 2023), FlowMM (Miller et al., 2024), FlowLLM (Sriram et al., 2024b), MatterGen-MP (Zeni et al., 2025), CrystalLLM (Gruver et al., 2024), and ADiT (Joshi et al., 2025). Our main results are presented in Table 1. On stability and S.U.N, the most important unconditional generation metrics, PLaID++ outperforms all prior methods. For our best overall PLaID++ model, 22.27% of the relaxed structures are deemed stable, out of which 72% are novel, and 55% are unique, leading to a S.U.N. rate of 7.74%. **PLaID++ achieves the highest stability rate and a ~50% higher S.U.N rate than the best previous method.**

On the proxy validity metrics, PLaID++ achieves performance on par with other methods. Our Wyckoff encoding specifically increases compositional validity for both the base and DPO fine-tuned models, highlighting symmetry-based encodings as a natural and intuitive mechanism to increase the validity and stability of generated structures. Though many of these metrics have saturated, we report

PLaID++'s performance for comparison and completeness.

We also observe that PLaID++ is significantly more computationally efficient than comparable methods. On a singular NVIDIA 80GB H100 GPU, PLaID++ samples 10,000 crystals in approximately 23 minutes, yielding 27.17 S.U.N. crystals per minute. **This generation speed is 5x faster than FlowLLM**, which takes 89.6 minutes to generate 10,000 materials on a NVIDIA 80GB A100 GPU, yielding 5.25 S.U.N. crystals per minute (Sriram et al., 2024b). Such high throughput is essential in real-world discovery, as faster sampling dramatically shortens the design–build–test cycle, enabling rapid screening of candidates and speedups of experimental pipelines (Szymanski et al., 2023; Curtarolo et al., 2013).

**Space Group Conditioned Generation** To test the ability of PLaID++ to generalize towards targeted structure synthesis, we measure the S.S.U.N. percentage for four model variants: (i) the 3D coordinate fine-tuned Qwen model (ii) the Wyckoff fine-tuned Qwen model, (iii) DPO with data only from the conditional generation task (iv) DPO trained jointly on conditional and unconditional generation data (flagship PLaID++). As shown in Figure **??**, applying DPO to space-group pairs increases S.S.U.N. by an average of 22% compared to the base Wyckoff model. Incorporating additional preference data from unconditional generation further boosts performance, resulting in a total increase of 47% over the base model—more than doubling the improvement from space-group conditioning data alone.

Groups with low training data occurrences ($<$ 400 samples each) like ($Amm2$, $Im2$, $P6_3/mmc$) show little improvement or sometimes slight degradation after DPO. This sup-

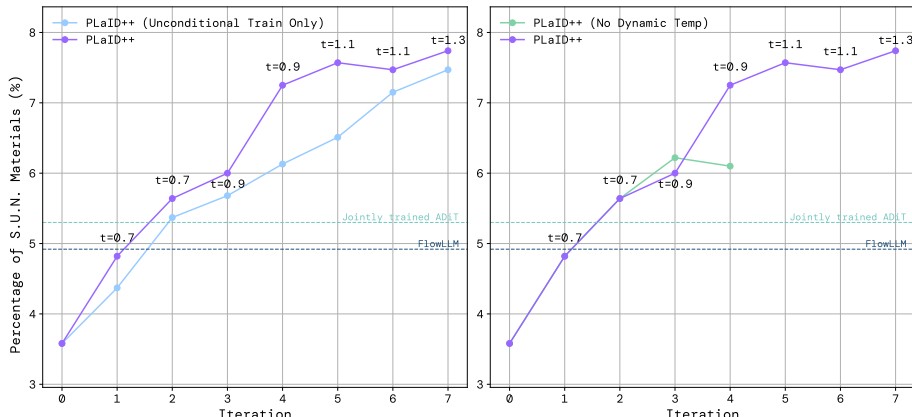

*Figure 3.* Evolution of S.U.N. percentage of PLaID++ variants' over DPO iterations. Reference lines represent S.U.N. rates from ADiT and FlowLLM's flagship models. **Left:** Ablation over joint training. **Right:** Ablation over dynamic temperature.

*Table 2.* Results across PLaID++ variants for Bulk Modulus (BM) (>325 GPa) conditioned generation across 1000 samples.

| Variant | High BM & S.U.N. | High BM |
|---|---|---|
| Wyckoff Base | 25 | 38 |
| PLaID++ (BM only) | 28 | 32 |
| **PLaID++ (Joint)** | **40** | **54** |

ports the elicitation hypothesis of reinforcement learning (Lambert, 2023), whereby RL amplifies existing valuable behaviors in the base model rather than teaching models new ones. With too few examples, neither the space-group nor the stability objective can generate a meaningful training signal for the model to learn from.

**Multi-Objective Property-Guided Generation** To test whether RLIP extends beyond stability and symmetry to arbitrary property targets, we add bulk modulus as a third preference objective, focusing on the superhard regime (>325 GPa). We construct (accept, reject) pairs from (stable & high-BM, stable & low-BM) and analogous tiers, with values predicted by eSEN (Fu et al., 2025). During training, we combine these pairs with our existing stability, novelty, and space-group preference sets under the same iterative DPO procedure (see Appendix 7.2 for full details). As shown in Table 2, joint preference optimization over all four objectives generates ~**40% more S.U.N. crystals satisfying our bulk modulus target** than RLIP on bulk modulus alone (40 vs. 28), and nearly 70% more high-bulk-modulus crystals overall (54 vs. 32). These results demonstrate how RLIP is a *general* procedure that can be applied whenever candidate materials can be ranked by a property evaluator, such as an MLIP or DFT.

**Density Functional Theory Performance Validation** We relax 1,000 unconditional generation structures from PLaID++ using DFT (see Appendix 7.2 for details). We

find a stability rate of approximately 19.1% and a S.U.N. rate of 13%. These results are consistent with our findings using the eSEN machine learning interatomic potential (Fu et al., 2025), providing strong empirical support for the quality of our generated materials. Notably, the DFT-based S.U.N. rate is higher than that obtained from the full 10,000-sample evaluation using eSEN. This discrepancy is expected due to the smaller DFT sample size, which yields a higher relative uniqueness rate.

## 5.4. Discussion

**Representation Choice is Critical to Prevent Mode Collapse**

In Table 3, we demonstrate the importance of our crystallographic representation. As we perform more iterations of DPO with the 3D coordinate representation, the diversity of the distribution quickly collapses and the model struggles to generalize from the DPO examples seen during training. Each iteration of the post-trained model has a progressively lower S.U.N. rate even as stability increases. This shows that **with the incorrect representation, RL leads the model to memorize rather than generalize from reward feedback.** We hypothesize that our Wyckoff representation works because of its higher information density, where local atom type or coordinate changes propagate to globally meaningful structure differences, expanding the diversity of valid structures the model can learn and generate. This is further reinforced by the ablation on the SFT base models in Table 1, where despite similar stability rates, the uniqueness and novelty of the Wyckoff representation is significantly higher, leading to a ~25% increase in S.U.N.

**Reinforcement Learning from Interatomic Potentials (RLIP)** Across multiple iterations of Direct Preference Optimization (DPO), we observe in Figure 3 that DPO successively improves the number of generated S.U.N. materi-

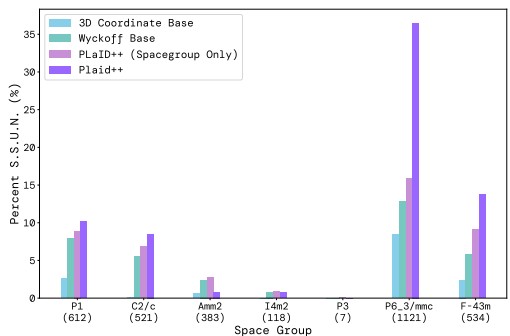 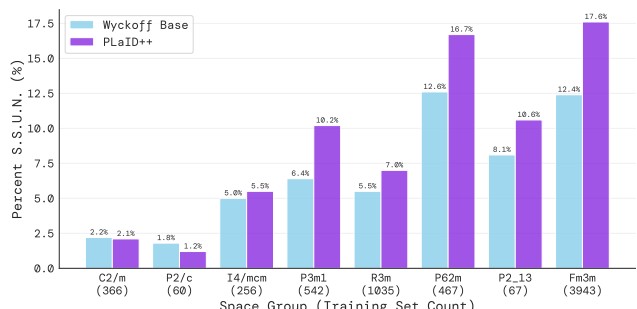

*Figure 4.* **Left:** Histogram of S.S.U.N. results across models for space-group conditional generation tasks. **Right:** S.S.U.N. rates for space-group–conditioned generation on space groups not directly optimized for in our RLIP pipeline; PLaID++ improves stability, uniqueness, and novelty across most space groups vs. the Wyckoff base model.

*Table 3.* PLaID++ 3D coordinate variant's S.U.N. and stability percentage across iterations of RLIP.

| Metric | It 1 | It 2 | It 3 |
|---|---|---|---|
| S.U.N. (%) | 2.93 | 2.23 | 1.69 |
| Stability (%) | 10.35 | 13.47 | 15.94 |

als, demonstrating that reinforcement learning is an effective and robust method to improve generation performance. On unconditional and space group conditioned generation, **we improve S.U.N. by ∼115% and ∼50% respectively compared to fine tuning alone**.

From our ablation on temperature sampling in Figure 3, we highlight that removing dynamic temperature updates across DPO iterations leads to mode collapse, where the S.U.N. rate degrades even as stability continues to increase after 3-4 iterations. This suggests that increasing temperature acts as an entropy regularizer which encourages exploration and counteracts model collapse.

**Joint Training Improves Performance**   Table 1 and Figure 4 show that our DPO model trained together on the unconditional and space group conditioned generation tasks outperforms the DPO models trained individually on each respective task. This validates the efficacy of a universal foundational model that can be leveraged across multiple downstream objectives. The performance gains are particularly pronounced in conditional generation, where only a few hundred samples are available in the training set. This suggests that the additional training signal derived from unconditional generation feedback can transfer to conditional generation.

**RLIP Generalizes Beyond Its Reward Structure**   In Figure 4, we also test PLaID++ on 8 space groups excluded from the set of space groups we directly optimized for in our RLIP pipeline. Our post-trained PLaID++ model sees

an average 20% improvement on SSUN compared to fine-tuning alone. This demonstrates that **reinforcement learning provides additional benefit to tasks not explicitly incentivized by our reward structure.** Notably, Figure 9 demonstrates that the improvements come from more than simply generating more stable materials—the proportion of materials that correctly satisfy the space group constraint also increases for many of the space groups. This highlights the benefits of a unified foundational model across many tasks, and suggests our RL approach learns generalizable structural principles that apply across space groups rather than merely memorizing training distribution patterns.

**RLIP Enables Local Exploration**   To better understand how reinforcement learning reshapes generative behavior, we visualize the compositional embedding space of S.U.N. crystals from the Wyckoff Base and PLaID++ models using UMAP (McInnes et al., 2018) (Figure 5). The left plot highlights a clear distributional shift: while both models occupy similar structural regions, PLaID++ expands into areas underrepresented in the base model, providing concrete evidence that RLIP enables local exploration around novel materials in our supervised baseline. In the right plot, we analyze the largest crystal cluster and observe that PLaID++ not only produces substantially more samples but also preferentially generates crystals containing P-block metalloids. Together, these visualizations demonstrate that PLaID++ is not merely memorizing preferences but uncovering out-of-distribution materials with meaningful structural diversity.

## 6. Conclusion

In this paper, we introduce a novel Wyckoff-based text encoding for crystal structures and introduce a new reinforcement learning framework, Reinforcement Learning from Interatomic Potentials (RLIP), to guide generation towards chemically stable and useful structures. Our method achieves state-of-the-art performance across stability and

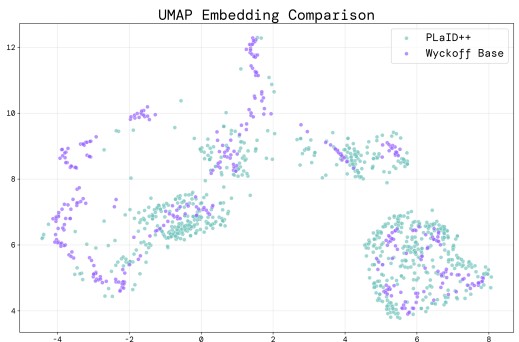 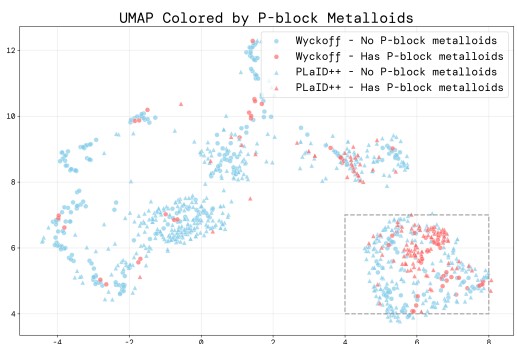

*Figure 5.* **Left:** UMAP visualizations of compositional embeddings of S.U.N. crystals generated by our PLaID++ and 3D Coord Base models. **Right:** Identical data, with crystals containing P-block metalloids being highlighted in red and those without P-block metalloids being in blue.

S.U.N. metrics. Current random structure search methods achieve less than a 1% success rate (Pickard and Needs, 2011) in identifying stable materials—PLaID++'s conditional and unconditional generation capabilities represent a significant acceleration from traditional approaches to future methods with greater real-world utility.

While PLaID++ achieves state-of-the-art stability and S.U.N. rates, there exist many promising avenues for future work. A natural next step would be to examine how performance scales with model and training set size on larger, more diverse datasets such as Alexandria (∼2.6M structures) (Schmidt et al., 2024) or LeMaterial (∼6.7M structures) (Siron et al., 2024). We also focused on Direct Preference Optimization (DPO) as our RL method due to the slow and compute-intensive nature of S.U.N. calculations; investigating alternative algorithms such as Proximal Policy Optimization (PPO) (Schulman et al., 2017) or Group Relative Policy Optimization (GRPO) (Shao et al., 2024) might uncover different stability–diversity trade-offs.

## Impact Statement

Our approach has the potential to accelerate the discovery of novel materials for renewable energy, electronics, and carbon-capture—paving the way for technologies that benefit society. However, deploying generative models carries risks, including synthesizing dangerous compounds and inequitable access to these tools.

## Acknowledgments

This work used high performance computing resources from Accelerating Computing For Emerging Sciences (ACES) at Texas A&M University through allocation CIS240657 from the Advanced Cyberinfrastructure Coordination Ecosystem: Services & Support (ACCESS) program, which is supported by U.S. National Science Foundation grants #2138259, #2138286, #2138307, #2137603, and #2138296. The authors also thank Modal for their compute grant that provided additional GPU compute during the peer review process.

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

# 7. Appendix

## 7.1. Proxy Metric Details

**Validity** provides a computationally efficient check on whether a generated crystal is physically plausible. We assess this through two criteria: *structural validity*, which ensures that no two atoms are closer than 0.5 Å, and *positional validity*, which verifies charge neutrality.

**Stability Rate** assesses how many generated materials are thermodynamically stable. Stability is determined by the energy above the hull ($E^{hull}$) metric, which measures the energy difference between a material and the convex hull of competing phases with the same composition. A material is considered stable if $E^{hull} \leq 0$ eV/atom, while those with $E^{hull} < 0.1$ eV/atom are classified as metastable. We compute $E^{hull}$ by relaxing generated structures using eSEN and the Materials Project dataset (February 2023) as a reference. Total energies were corrected using the MP2020 compatability scheme, which maintains consistency across various functionals (DFT/DFT+U).

**S.U.N. Rate** extends the stability metric to assess both novelty and uniqueness. A structure is novel if it is not structurally similar to any training set material, determined using Pymatgen's StructureMatcher (Ong et al., 2013). Uniqueness ensures that duplicate generations are not counted separately by grouping structurally similar outputs into equivalence classes. The S.U.N. rate is defined as the fraction of generated structures that are Stable, Unique, and Novel:

$$\text{Stability Rate} = \frac{N_{\text{stable}}}{N_{\text{gen}}} \times 100\% \qquad (2)$$

$$\text{SUN Rate} = \frac{N_{\text{SUN}}}{N_{\text{gen}}} \times 100\% \qquad (3)$$

Together, these metrics provide a more rigorous assessment of generative model performance by quantifying both the stability and originality of discovered materials.

## 7.2. Additional Experiment Details

**Wyckoff Crystal Representation** We represent each crystal structure in PLaID++ as a structured text sequence that encodes key physical and symmetry-related attributes. The representation begins with the chemical formula, expressed as a concatenation of each element and its corresponding multiplicity. This is followed by the space group number, determined using Pyxtal (Fredericks et al., 2021) with a symmetry tolerance of 0.01. Next, we include the lattice parameters: side lengths and angles, each rounded to two decimal places. Finally, for each atom in the structure, we list the element type, fractional coordinates (rounded to three decimal places), Wyckoff site label, and site multiplicity also from Pyxtal (Fredericks et al., 2021). This format provides a comprehensive and consistent description of a crystal's composition, geometry, and symmetry. We provide additional examples of generated structures in Appendix 7.9.

**Supervised Finetuning** We perform supervised finetuning (SFT) on Qwen to adapt the model for crystal structure generation. For our fine-tuning, the model performs infilling one-third of the time. The remaining two-thirds of the time, we generate structures *de novo*, where we either generate crystals unconditionally, or randomly append between one and five additional properties in our prompt. The properties we include are the chemical formula, energy above the hull, formation energy per atom, band gap and space group although space group generation was the focus of this work.

To fine-tune different Qwen-2.5 7B models on both text-based and Wyckoff-based crystal representation, we followed the steps outlined by Gruver et al. (2024) to reproduce their results. We use an AdamW optimizer with batch size of 16 samples, a learning rate of $10^{-5}$, fp-4 mixed precision alongside LoRA adapters (with a LoRA rank of 8, LoRA alpha of 32, and LoRA dropout of 0.05) to fine-tune over the MP-20 dataset for 10 epochs.

**Iterative SFT Ablation** To directly compare rejection-sampled supervised finetuning (SFT) against Direct Preference Optimization (DPO), we trained a baseline that repeatedly fine-tuned only on metastable structures, matching our DPO iteration count, learning rate, and sample size. Results are summarized below.

We find that although iterative SFT improves upon vanilla SFT, it underperforms DPO by a large margin, underscoring the benefit of fine-grained preference learning for stability and novelty.

**DPO** After supervised finetuning, we apply Direct Preference Optimization (DPO) to align the model's probability distribution to favor selected structures. We use the DPO Trainer from TRL (Wolf et al., 2020) using the Adam optimizer with a batch size of 16 samples, a learning rate of $10^{-6}$, fp-4/bfloat-16 precision, and a $\beta$ of 0.1 on one epoch of our dataset.

To fine-tune our model using Direct Preference Optimization (DPO), we curate a dataset of preference pairs drawn from two sources: (1) unconditional generation outputs and (2) space group–conditioned generation outputs. These datasets are merged and used jointly in the DPO training objective.

For the 10,000 samples generated from our unconditional generation task, we classify each structure using energy above the hull as predicted by eqV2. We use the following stability thresholds: stable ($E^{hull} \leq 0$), metastable ($0 < E^{hull} \leq 0.08$), and unstable ($E^{hull} > 0.08$). We form preference pairs using the following procedure. For each stable crystal, we sample one metastable and two unstable

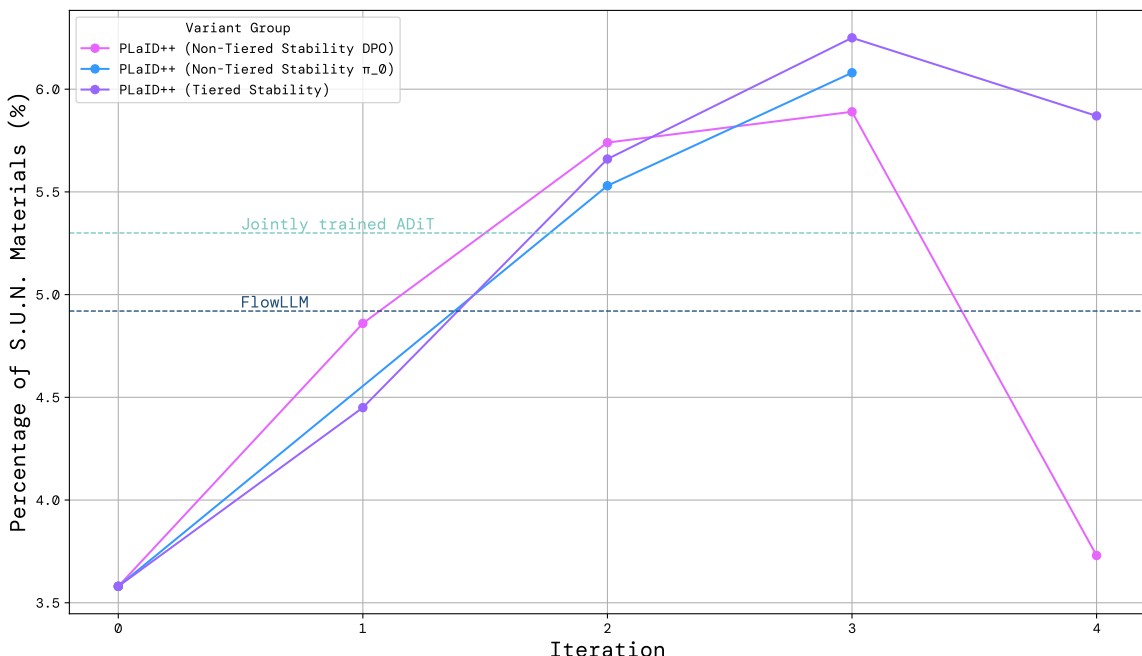

*Figure 6.* DPO Ablation, highlighting percentage of S.U.N. Materials across iterations of PLaID++ variants with different DPO pairs

crystals to create three pairs: one (stable, metastable) and two (stable, unstable). For each metastable crystal, we sample two unstable crystals to create (metastable, unstable) pairs.

This tiered pairing strategy provides a more fine-grained reward signal than binary stable/unstable classification and avoids overfitting to exact energy values while increasing the diversity of preference samples.

We also construct a preference dataset from crystals generated using prompts that specify a desired space group number as described in Figure 2. For each of the seven selected space groups, we sample 1,000 structures and compute both the eqV2-predicted energy and the actual space group using Pyxtal with a symmetry tolerance of 0.01 (Fredericks et al., 2021). Crystals with $E^{\text{hull}} \leq 0.08$ are considered acceptable. Within this set, we further divide crystals into those that match the target space group and those that do not. We construct preference pairs by sampling (matching SG, non-matching SG) pairs among stable/metastable crystals and sampling (stable/metastable and unstable) pairs in a 1:2 accept reject ratio to enforce both symmetry and stability constraints.

By combining these unconditional and conditional preference pairs into a single dataset, we enable the model to jointly learn to generate stable crystals and adhere to symmetry constraints within a unified DPO training pipeline.

This creates between 10,000 to 20,000 DPO pairs depend-

ing on the iteration number and metastability and symmetry rates. Interestingly, the model incurs worse performance when we compile our dataset using a 1:1 and 1:10 ratio. We hypothesize that smaller ratios perform worse as our training dataset becomes noticeably smaller, and larger ratios perform worse as the model overfits to each positive sample and consequently generates less diverse structures and lower S.U.N. rates.

**Bulk Modulus Ablation Details**     This appendix expands on the multi-objective property-guided generation experiment in the main paper.

Bulk modulus values are predicted using our eSEN MLIP. We define *high bulk modulus* crystals as those with predicted bulk modulus between 325 and 725 GPa, restricting the upper bound to avoid outlier crystals that may reflect reward hacking of the MLIP rather than valid bulk modulus predictions. Prior work has shown that MLIPs align well with ground-truth DFT-calculated bulk modulus values, providing confidence that the model is generating crystals with realistic mechanical properties.

We introduce additional (accept, reject) preference pairs consisting of (stable & high bulk modulus, stable & low bulk modulus), (stable & novel & high bulk modulus, stable & novel & low bulk modulus), and (stable & high bulk modulus, unstable & high bulk modulus) crystals. These pairs are combined with the existing stability, novelty, and space-group-based preference datasets and optimized jointly using the same iterative DPO procedure as in the main paper.

*Table 4.* Band-gap-targeted S.U.N. generation counts. Columns report the number of generated structures that are S.U.N. and exceed each band gap threshold.

| Model | > 1 eV | > 2 eV | > 3 eV |
|---|---|---|---|
| Wyckoff Base (Non-DPO) | 4 | 1 | 0 |
| PLaID++ (It 1) | 15 | 9 | 3 |

**Band Gap Ablation Details**    To provide additional evidence that our pipeline is a general procedure, we ran an additional experiment extending PLaID++ to targeted band gap generation. The band gap is a fundamental electronic property controlling whether a material behaves as a metal, semiconductor, or insulator, making it central to applications such as photovoltaics, LEDs, and power electronics. In particular, we use the dedicated property-prediction GNN MEGNet (Chen et al., 2019) to predict band gaps. We construct tiered preference pairs analogous to our bulk modulus pipeline, ranking crystals by high band gap ( $\geq$ 3.0 eV), moderate band gap ($\geq$ 1.0 eV, < 3.0 eV), and low band gap, in combination with stability and novelty. Even after a single RLIP iteration, PLaID++ shows clear improvement over the non-DPO baseline across 1000 generated samples in Table 4. These results reinforce that our RLIP pipeline can be successfully applied across a range of crystal properties.

**DPO Ablation**    We analyze the S.U.N. performance for different DPO variants in Figure 6. The PLaID++ (Non-Tiered Stability DPO) variant applies DPO using stability-based preference pairs, where accepted crystals are stable or metastable and rejected crystals are unstable. The PLaID++ (Non-Tiered Stability $\pi_0$) variant keeps the reference model fixed as the supervised fine-tuned Qwen Wyckoff base model instead of updating the reference to the previous iteration's model ($\pi_{\text{ref}} = \pi_{\theta-1}$) as in the flagship PLaID++ method. The PLaID++ (Tiered Stability) variant is the model which uses tiering on (stable, metastable), and (metastable, unstable) pairs. The results demonstrate the efficacy of both using a tiered DPO stability objective and using the previous iteration model as the reference model for KL divergence calculation.

**eSEN-30M-OAM and eqV2 MLIPs**    We employ eqV2 to curate our preference dataset (assessing crystal structures above and below the 0.08 eV/atom $E^{\text{hull}}$ threshold) and eSEN-30M-OAM (eSEN) for our stability evaluations by computing relaxed formation energies. For eqV2 and eSEN, we used their reported settings with 500 relaxation steps and a max force of 0.02 (Fu et al., 2025; Barroso-Luque et al., 2024).

**Density Functional Theory Setup**    We used the Vienna Ab initio Simulation Package (VASP) version 6.3.2 software (Kresse and Hafner, 1993), (Kresse and Furthmüller, 1996), (Kresse and Furthmüller, 1996). All unit cells were relaxed

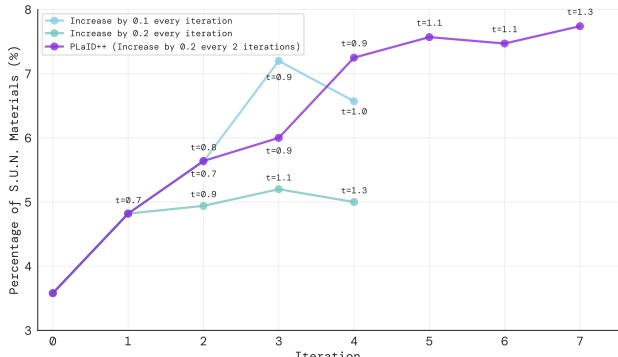

*Figure 7.* Comparison of temperature schedules during DPO. Fast temperature ramps (0.1/iter, 0.2/iter) degrade S.U.N. performance, while the PLaID++ schedule (0.2 every two iterations) balances the exploration-exploitation tradeoff, yielding the best results.

using the default VASP parameters used for calculations in the Materials Project database as described in Pymatgen v2025.5.1 (Ong et al., 2013). These are collinear spin-polarized calculations which include a plane wave energy cutoff of 520 eV, a maximum force tolerance of $0.5 \times 10^{-5}$ eV/Å, and a $\Gamma$ centered k-point mesh of 64 k-points per Å$^3$.

**Experiments – Server Details**    We conducted training on a high performance computing (HPC) cluster equipped with NVIDIA H100 GPUs. We used a single NVIDIA H100 GPU for Qwen-2.5 7B SFT and DPO as well as inference. We used an AMD Epyc 7443P for S.U.N. rate evaluations and DFT calculations.

### 7.3. Wyckoff Representation Scaling Analysis

A natural concern with our Wyckoff representation is whether its token savings degrade for low-symmetry structures, where atoms may increasingly occupy distinct sites and the encoding approaches a per-atom listing. To investigate this, we sampled a subset of MP-20 crystals from each crystal system and tokenized them under both the 3D coordinate and Wyckoff encodings using PLaID++'s Qwen-2.5 tokenizer.

*Table 5.* Average token counts per crystal under 3D coordinate and Wyckoff encodings on a subset of MP-20, stratified by crystal system. Wyckoff encodings save tokens across all symmetry classes, with the largest gains at moderate-to-high symmetries.

| Symmetry | Coord | Wyckoff | Reduction |
|---|---|---|---|
| Low (Monoclinic) | 211 | 174 | 17% |
| Moderate (Ortho./Tetragonal) | 230 | 147 | 36% |
| High (Cubic/Hexagonal) | 214 | 145 | 32% |
| **Overall** | **218** | **155** | **29%** |

Wyckoff encodings save tokens across every symmetry class, with reductions exceeding 30% for moderate and high-

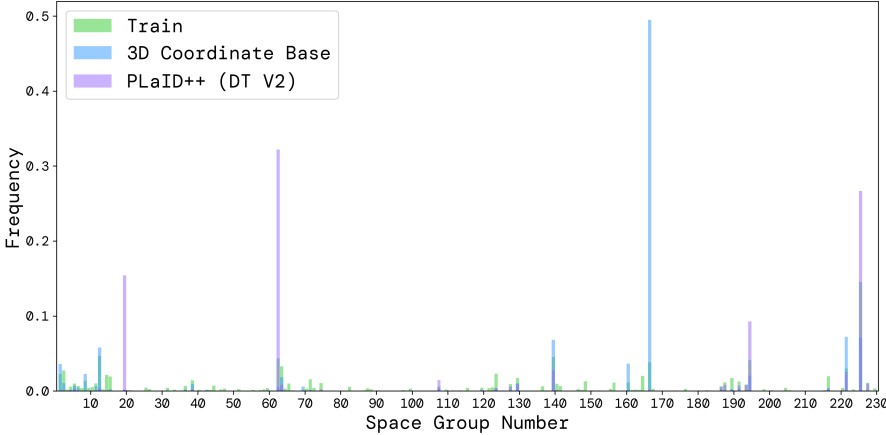

*Figure 8.* A comparison between the three space group distributions allows us to extract valuable insights on how our PLaID++ techniques affect model behavior.

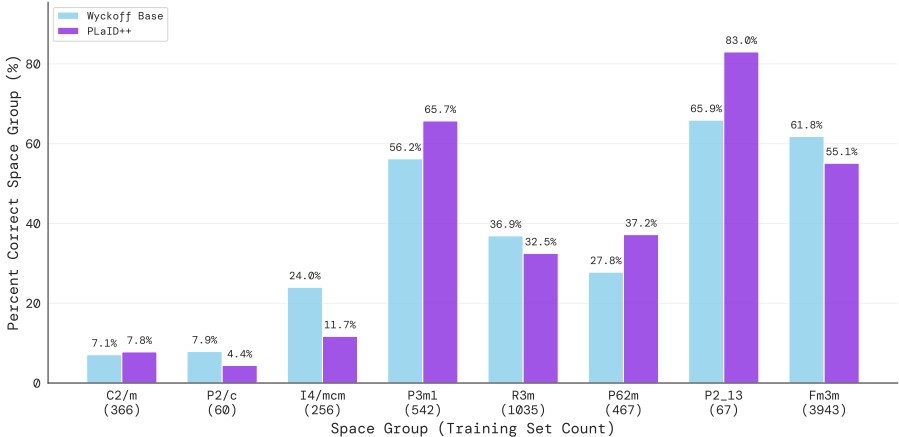

*Figure 9.* Space-group accuracy for conditioned generation on space groups not directly optimized for in our RLIP pipeline. PLaID++ generates more crystals in the correct space group for most groups.

symmetry structures. The savings shrink for low-symmetry crystals because higher-multiplicity sites become rare: in the worst case of $P1$, the Wyckoff encoding degenerates to a per-atom listing equivalent in length to the coordinate baseline. Since the vast majority of known inorganic materials exhibit nontrivial symmetry, Wyckoff remains a robust default in practice.

### 7.4. External Benchmark Validation

The recently released LeMat-GenBench evaluates 12 recent crystal generative models using a unified protocol with LeMat-Bulk as the reference set, a multi-MLIP ensemble for energy evaluation, and standardized validity, stability, uniqueness, novelty, and S.U.N. metrics (Betala et al., 2026). This benchmark is substantially stricter than MP-20-based evaluation as LeMat-Bulk contains roughly 5 million structures, producing a tighter convex hull and a more comprehensive novelty reference. Under this setting, PLaID++

achieves the highest Stable% and S.U.N.% among the evaluated generative models, producing S.U.N. crystals at over $2\times$ the next best model. This external validation supports our central claim that RLIP substantially improves discovery rates, even under a broader and more stringent evaluation protocol.

Importantly, these results contextualize the trade-offs observed throughout our paper. LeMat-GenBench reports that no model dominates all dimensions: models such as Matter-Gen and WyFormer retain higher novelty and uniqueness, while PLaID++ is more strongly optimized for thermodynamic stability and stable-unique-novel generation. This is consistent with the design of our method, which does not aim to maximize unconstrained exploration, but instead uses preference alignment to shift high-throughput generation toward physically useful regions of chemical space. PLaID++'s performance as the best model on Stable% and S.U.N.% under an independently designed benchmark fur-

| Model | Stable% ↑ | S.U.N.% ↑ |
|---|---|---|
| MatterGen | 2.0 | 0.2 |
| PLaID++ | **12.4** | **1.0** |
| WyFormer | 0.5 | 0.1 |
| WyFormer-DFT | 3.7 | 0.4 |
| ADiT | 0.4 | 0.0 |
| DiffCSP | 2.3 | 0.1 |
| DiffCSP++ | 1.0 | 0.2 |
| SymmCD | 1.4 | 0.1 |

*Table 6.* Compressed LeMat-GenBench stability results using LeMat-Bulk as the reference set and an MLIP ensemble for evaluation. PLaID++ achieves the highest stability and S.U.N. rates among the selected generative models.

ther supports that the gains observed in our experiments are not merely due to our reward design, but reflect an advantage for physically-informed crystal discovery that can generalize across reference sets and evaluation methods.

### 7.5. Space Group Distribution Analysis

We compare visualizations between the space group distribution of the MP-20 dataset we used to perform SFT on our base models, a sample of 10,000 crystals from the Qwen-2.5 7B SFT model trained on the 3D coordinate representation, and a sample of 10,000 crystals we from our PLaID++ model trained on iterative DPO finetuning with our Wyckoff representation in Figure 8. We note that the flagship model favors generating out-of-train-distribution space groups such as $P2_12_12_1$ (19) and $Pnma$ (62), a consequence of our RLIP framework's novelty incentive.

Space groups $Cm$ (8) and $P1$ (1) are prominent in the base 3D coordinate representation PLaID++ model's crystals, but not in the flagship model. Space group *Cm* has a single symmetry operation (a glide plane), while space group $P1$ only has the identity operation. In addition, space groups $P6_3/mmc$ (194) and $I4mmm$ (139) are among the five most frequently generated space groups in the flagship model, even though they are not in the top five of the training set. This supports the hypothesis that the Wyckoff representation helps PLaID++ generate systems with complex symmetry requirements, skewing the distribution towards generating crystals with higher symmetries and ensuring a more even distribution across all space groups.

### 7.6. Validation of MLIPs as a Surrogate for DFT Energy Above Hull

To justify using eSEN and eqV2 MLIP energies for evaluations in place of expensive DFT calculations, we compared eSEN/eqV2-predicted and DFT-computed energies above the convex hull ($E_{hull}$) on a sample of 1000 generated structures. All structures were sampled from PLaID++, then fil-

tered to those whose DFT energies are between $[-0.2, 0.2]$ eV/atom. This "near-hull" window was chosen because (i) our stability metric depends on correctly classifying structures around the zero-energy threshold, (ii) crystals with $|E_{hull}| > 0.2$ eV/atom either fail catastrophically in DFT or are unambiguously unstable/stable, making the precise MLIP $E_{hull}$'s accuracy less relevant for our evaluation metrics like S.U.N, and (iii) correctly classifying crystals as stable, metastable, or unstable is essential for creating accurate preference pairs used during the iterative DPO process.

**eSEN**. Figure 10 on the left shows a strong linear correlation ($R^2 = 0.84$) between eSEN and DFT energies in the critical near-hull region. Based of this high coefficient of determination, eSEN serves as a reasonable proxy for DFT in our evaluation pipeline: it reproduces stability classifications (stable vs. unstable) and catastrophic failure cases outside $[-0.2, 0.2]$ eV/atom while enabling a large reduction in computational cost. Experimentally, eSEN relaxes 10,000 crystals in approximately 5 hours, whereas DFT requires 12 hours for 1,000 crystals—yielding a ∼24× speedup with only a minimal accuracy trade-off.

We calculate eSEN's accuracy as a proxy for DFT in evaluation of the stability and metastability of crystals. When classifying the stability of structures at 0.0 eV eSEN achieves a precision of 0.696, a recall of 0.688 and $F_1 = 0.692$ (accuracy = 0.90), highlighting its accuracy in stability classification. Allowing for a small metastability margin (cutoff = 0.1 eV) boosts precision to 0.933 and recall to 0.992, giving $F_1 = 0.962$ (accuracy 0.95). Although imperfect, eSEN's relatively high F1 and accuracy scores demonstrate that it can be useful to approximate metrics like stability rate and S.U.N. without needing the prohibitive cost of DFT on a full 10,000 samples.

**eqV2**. Figure 10 demonstrates a strong linear correlation ($R^2 = 0.68$) between eqV2 and DFT energies near the stability threshold. Although noisier than eSEN, its high coefficient of determination demonstrates that eqV2 is a reliable surrogate for DFT in the critical near-hull region: replacing DFT with eqV2 in reward evaluation allows the pipeline to accurately create preference pairs for the iterative DPO process at a significantly lower computational cost compared to DFT. Experimentally, eqV2 performed at similar speeds to eSEN, yielding a ∼25× speedup over DFT.

We further justify eqV2's ability to act as an accurate and efficient reward model proxy for the crystal stability and metastability classification tasks. At the stability cutoff (0.0 eV/atom), eqV2 attains a precision of 0.591, a recall of 0.818, and an $F_1$ score of 0.686 (accuracy = 0.87), reflecting strong sensitivity but moderate specificity around the exact hull threshold. Increasing the cutoff to metastability (0.08 eV) boosts precision to 0.915 and recall to 0.992, yielding an $F_1$ score of 0.952 (accuracy = 0.94). This justifies our

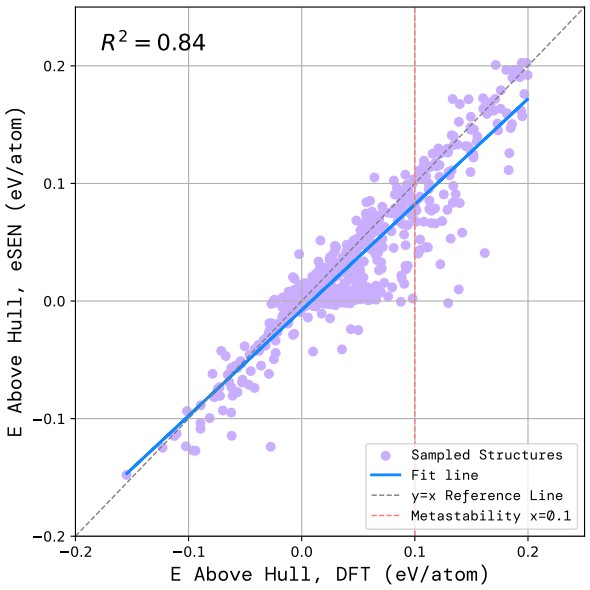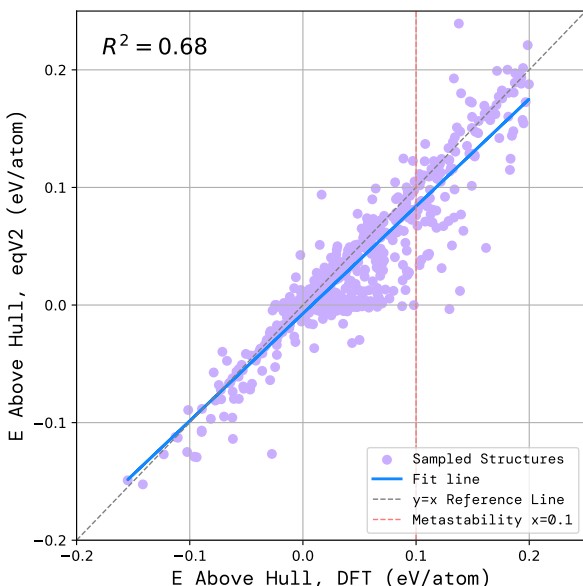

*Figure 10.* Scatter of MLIP vs. DFT energies above hull for 1000 sampled structures each with a least-squares fit line. **Left:** eSEN predictions ($R^2 = 0.84$). **Right:** eqV2 predictions ($R^2 = 0.68$).

tiered DPO reward formulation, whereby we provide the model with useful but noisy stability information and more accurate but less useful metastability information.

### 7.7. PLaID++ Overview

Algorithm 1 provides an overview of the full PLaID++ pipeline for the unconditional case (without space-group conditioning). Note that $s_w \succ s_l$ denotes preference (stable $\succ$ metastable $\succ$ unstable).

### 7.8. Wyckoff Theory: Mathematical Background

A Wyckoff position defines a set of equivalent atomic sites in a given space group (Evarestov and Smirnov, 1997). These positions are characterized by symmetry constraints, which reduce the degrees of freedom in atomic placement.

For a crystal structure belonging to a space group $G$, the Wyckoff positions can be defined as:

$$W = \{gx | g \in G\} \tag{4}$$

where $x$ is a fractional atomic coordinate, and $g$ is an element of the space group.

Each Wyckoff site follows symmetry constraints that ensure atoms occupy positions dictated by the underlying crystallographic symmetry:

$$x' = Rx + t \tag{5}$$

where $R$ is a symmetry operation (such as a rotation, reflection, or inversion) that can be represented as a matrix, and $t$ is a translation vector.

In this notation, $x'$ is considered to be part of the same crystallographic orbit as $x$, where $x$ is referred to as the generating point. As any one arbitrary point in the crystallographic orbit can be used to generate all other points in the orbit, the choice of $x$ is not unique.

All symmetry operations $(R, t)$ which map a point in the crystallographic orbit to itself form a finite subgroup of $G$, which can be identified as one of the 32 crystallographic point groups. Each Wyckoff site in each space group is associated with one such point group, and is often labeled according to the multiplicity of $W$ (the number of points in the orbit generated by the space group in a single unit cell) and a letter encoding the point group (i.e. '4a').

### 7.9. Miscellaneous Examples of Generated Crystals

We present additional qualitative examples of crystal structures generated by our model. The prompts and generated Wyckoff representations from PLaID for both unconditional generation and two examples of space group conditioned generation are shown in Table 7. Seperately, we visualize four crystals using the ASE and PyMatGen toolkits. Figure 11 shows several samples from different space groups.

---

**Algorithm 1** PLAID++ Pipeline

---

**Input:** Crystal corpus $\mathcal{D} = \{\hat{y}_i\}_{i=1}^{D}$          ▷ `In Wyckoff format`
  Initial policy $\pi_{\theta_{\text{ref}}}$          ▷ `QWEN-2.5 7B`
  Stability predictor $f_{\text{MLIP}}$          ▷ `eqV2 MLIP`
  Batch size $B$, DPO Regularization $\beta$, Prompt $x$
**Output:** PLAID++ policy $\pi_{\theta_i}$

**Supervised fine-tuning** $\theta_0 \leftarrow \theta_{ref}$          ▷ `Initialize from reference`
**for** mini-batch $\mathcal{B} = \{\hat{y}_b\}_{b=1}^{B} \sim \mathcal{D}$ **do**
     $\mathcal{L}_{SFT} = -\sum_{b=1}^{B} \log \pi_0(\hat{y}_b \mid x)$
       $\theta_0 \leftarrow \texttt{OptimizerStep}(\mathcal{L}_{SFT};\, \theta_0)$

**RLIP fine-tuning (Stability-only)** **for** $i = 1, \ldots, R$ **do**
     $\theta_i \leftarrow \theta_{i-1}$          ▷ `Initialize from previous round`
     $\{y_j\}_{j=1}^{K} \sim \pi_{\theta_{i-1}}(y \mid x)$          ▷ `Generate candidates`
     $s_j \leftarrow f_{\text{MLIP}}(y_j)$          ▷ `≈ e above hull`
     Create preference pairs $\mathcal{P} = \{(y_w, y_l)_j\}_{j=1}^{S}$ where $s_w \succ s_l$ according to stability/metastability thresholds
     **for** mini-batch $\mathcal{B} = \{(y_w, y_l)_b\}_{b=1}^{B} \sim \mathcal{P}$ **do**
         $\mathcal{L}_{\text{DPO}} = -\frac{1}{B}\sum_{b=1}^{B}\left[\sigma\!\left(\beta \log \frac{\pi_{\theta_i}(y_w|x)}{\pi_{\theta_{i-1}}(y_w|x)} - \beta \log \frac{\pi_{\theta_i}(y_l|x)}{\pi_{\theta_{i-1}}(y_l|x)}\right)\right]$
         $\theta_i \leftarrow \texttt{OptimizerStep}(\mathcal{L}_{DPO};\, \theta_i)$

**return** $\pi_{\theta_i}$          ▷ `Optimized policy`

---

*Table 7.* Example PLaID++ generations in response to unconditional and space group–conditioned prompts using the Wyckoff-based representation.

| Prompt | Model Output |
|---|---|
| `Below is a description of a bulk material.`
`Generate a description of the lengths and angles`
`of the lattice vectors and then the element type`
`and coordinates for each atom within the lattice:` | `Ho4Al4Ni4`
`Spacegroup:  Pnma`
`abc:  6.91 4.32 7.44`
`angles:  90.00 90.00 90.00`
`Sites (12)`
`Ho 0.005 0.250 0.195 4c`
`Al 0.190 0.250 0.590 4c`
`Ni 0.789 0.250 0.591 4c` |
| `Below is a description of a bulk material.  The`
`spacegroup number is 15.  Generate a description`
`of the lengths and angles of the lattice vectors`
`and then the element type and coordinates for each`
`atom within the lattice:` | `Mn8O12F4`
`Spacegroup:  C2/c`
`abc:  9.55 4.98 6.95`
`angles:  90.00 96.49 90.00`
`Sites (24)`
`Mn 0.826 0.495 0.074 8f`
`O 0.832 0.906 0.599 8f`
`O 0.000 0.489 0.250 4e`
`F 0.000 0.000 0.000 4a` |
| `Below is a description of a bulk material.  The`
`spacegroup number is 216.  Generate a description`
`of the lengths and angles of the lattice vectors`
`and then the element type and coordinates for each`
`atom within the lattice:` | `Ti4Ni4Sn4`
`Spacegroup:  F-43m`
`abc:  5.89 5.89 5.89`
`angles:  90.00 90.00 90.00`
`Sites (12)`
`Ti 0.500 0.500 0.500 4b`
`Ni 0.750 0.750 0.750 4d`
`Sn 0.000 0.000 0.000 4a` |

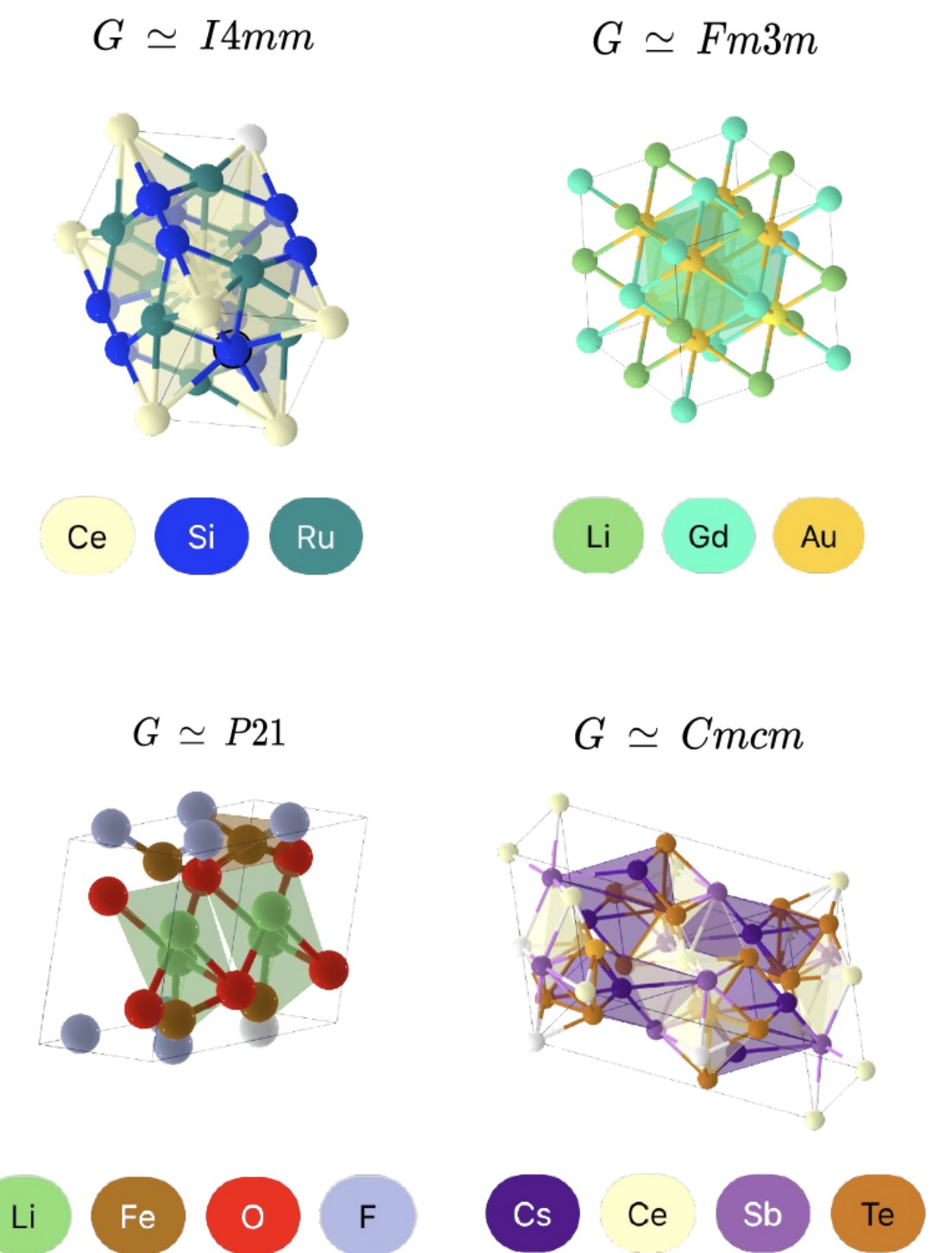

*Figure 11.* Examples of crystals generated by our Qwen-2.5-7B model fine-tuned with Wyckoff representation and Direct Preference Optimization (DPO). The crystals' corresponding chemical formulas are from top left to bottom right respectively, $Ce_2Si_4Ru_4$ , $Li_4Gd_4Au_8$, $Cs_4Ce_4Sb_4Te_{12}$, and $Li_4Fe_4O_4F_4$, respectively.

