# OpenReview forum: "PLaID++: A Preference Aligned Language Model for Targeted Inorganic Materials Design"
_ICML.cc/2026/Conference — ICML 2026 regular_

### Official Review · Reviewer_v4NK · 2026-03-08

**Soundness:** 3
**Presentation:** 3
**Significance:** 3
**Originality:** 4
**Overall Recommendation:** 5
**Confidence:** 3

**Summary:**

The authors combine reinforcement learning with preference based optimization to generative models for inorganic chemistry. PLaID++ is a post trained LLM that can generate stable and diverse crystal structures. The highlight of the paper is use of  symmetry-informed Wyckoff textual representations, curated dataset of stability crystal pairs for the. DPO framework and joint optimization on stability, novelty and diversity as objectives. Experiments show improvements across all three metrics and at a significantly higher rate.

**Compliance With Llm Reviewing Policy:**

Affirmed.

**Key Questions For Authors:**

Following are the questions:

1) Since MLIP is used for creating pairs, how sensitive is the result to the choice of the model? 2) Can the Wyckoff approach address issues in disordered structures? 3) Are the results better with increase in model sizes and if so is the stability an issue with increase in size? 4) Is any experimental validation done on the generated crystals?

**Limitations:**

1. Pairs created with only one model (MLIP) and the optimization may amplify those biases
2. Reward design can introduce bias, did not see it was discussed
3. Is 45k structure enough for LLM?

**Strengths And Weaknesses:**

The authors are able to draw ideas from different domains and propose their model on inorganic crystals. This is a great example of cross-domain contribution. While the DPO/ RLHF alignment with LLM is not novel but combination of several components towards generative modeling for materials is a great contribution. In addition they emphasize and improve several components: 1) the physical aspects of symmetry, 2) careful data curation based on MLIP relaxations such that the model generates structures with lower predicted energies, 3) token reduction and avoiding mode collapse as seen in prior work 4) increase throughput which is critical for practical applications. Finally a suite of benchmark modes are compared and the proposed model shows improvements across the three metrics. The ablations across components show gains from each of them.

---

> ### Author Rebuttal · Authors · 2026-03-31
>
> We thank the reviewer for their thoughtful review! We appreciate the recognition of the novelty and rigor of our cross-domain contributions.
>
> >_**Since MLIP is used for creating pairs, how sensitive is the result to the choice of the model?**_
>
> Great question! We have rerun our PLaID++ unconditional generation results on the Orb-V3 conf-inf-mpa MLIP introduced in [1]. On this MLIP, we obtain a 25.2% stability rate and 8.3% S.U.N. rate, slightly higher than the 22.27% stability and 7.74% S.U.N. rate evaluated on eSEN that was reported in the paper. This supports that our model is not specifically hacking to one MLIP or MLIP family.
>
> [1] Orb-v3: atomistic simulation at scale. Rhodes et al. arXiv 2025
>
> >_**Can the Wyckoff approach address issues in disordered structures?**_
>
> In crystallography, disorder is naturally expressed through fractional occupancies at a site (e.g., a site being 29% Ni and 71% Fe). Since our Wyckoff-based encoding already models atomic sites and symmetry explicitly, incorporating disorder would be straightforward by augmenting each site with occupancy weights (e.g., a distribution over elements), preserving the same symmetry-aware structure used throughout the paper.
>
> >_**Are the results better with increase in model sizes?**_
>
> Due to computational restrictions, we do not explore using other LLM base model sizes more than 7B parameters. Similar to work in the protein language model space [1], we expect the PLaID++ methodology in the materials domain to scale favorably with larger datasets and model sizes, and plan to investigate this in future work.
>
> [1] Scaling Unlocks Broader Generation and Deeper Functional Understanding of Proteins. Bhatnagar, A., et al. NeurIPS, 2025.
>
> >_**Is any experimental validation done on the generated crystals**_
>
> We validate 1,000 random samples with DFT relaxations with our PLaID++ flagship model, obtaining an improved 19.1% stability rate and 13% S.U.N. rate, well-aligned with the 22.27% and 7.74% eSEN-based rates over 10k generations. We also selected ~50 S.U.N. crystals for manual review from an expert crystallographer, who did not find any specific unphysical properties of our generations. Generating over 27 S.U.N. crystals per minute, PLaID++ is designed for producing a large volume of property-aligned, physically plausible candidates. To facilitate experimental validation, we will make all generated crystals across unconditional, space group conditioned, and high bulk modulus experiments, along with our model weights, publicly available.
>
> >_**Pairs created with only one model (MLIP)...**_
>
> We’d like to clarify that we train our DPO model using eqV2 as the reward signal, but evaluate final generations using eSEN (a separate MLIP), to reduce the likelihood of reward hacking. We agree that reliance on MLIP-derived preference pairs raises the question of whether PLaID++ might align to a family of interatomic potentials rather than true DFT or experimental behavior. To directly assess this, we performed an MLIP–DFT correlation study in Appendix 7.4 and found that eSEN predictions achieve $R^2=0.84$ when regressed against DFT. This strong correlation indicates that modern MLIPs provide an accurate and smooth approximation of DFT for constructing preference signals.
>
> >_**Reward design can introduce bias**_
>
> We agree that reward design is a critical source of potential bias, and we designed RLIP explicitly to mitigate this by avoiding brittle, scalar reward optimization in favor of relative, multi-objective preference learning. Instead of training directly on noisy MLIP outputs, we construct tiered preference pairs (stable > metastable > unstable), which encode ordinal relationships that are significantly more robust to MLIP noise. This allows the model to learn reliable gradients toward physically meaningful regions of chemical space without overfitting to specific energy values. We further decouple objectives by incorporating novelty as a separate preference signal ((stable & novel) ≻ (stable & not novel)), while uniqueness is controlled implicitly through temperature scaling, which we show in our main paper preserves distributional diversity during iterative DPO.
>
> Importantly, as we’ve demonstrated with bulk modulus, this framework is modular: any additional property (e.g., band gap) can be incorporated as another preference axis without redefining the objective, making the reward design extensible.
>
> >_**Is 45k structure enough for LLM?**_
>
> We’d like to clarify that the effective training signal in PLaID++ extends beyond the 45k MP-20 structures: through RLIP, the model learns from an additional ~120k model-sampled crystals via iterative preference feedback, yielding a substantially larger and continually refreshed dataset (i.e., self-generated supervision). More importantly, a key motivation of our work is to demonstrate that using RL, LLM-based generators can **operate effectively in data-sparse scientific regimes**.

---

> > ### Author Rebuttal · Reviewer_v4NK · 2026-04-01
> >
> > Thank you for addressing the concern. I maintain my score

---

### Official Review · Reviewer_fnZy · 2026-03-13

**Soundness:** 2
**Presentation:** 3
**Significance:** 3
**Originality:** 3
**Overall Recommendation:** 4
**Confidence:** 4

**Summary:**

The paper introduces PLaID++, a framework that treats inorganic crystal materials design
as a text generation task utilizing a Large Language Model. Instead of
using direct 3D coordinate representations, the authors propose a compact, symmetry-
informed text representation based on Wyckoff positions. The
RLIP framework aligns the model toward generating Stable, Unique, and Novel (S.U.N.)
materials by utilizing Machine Learning Interatomic Potentials (MLIPs) to construct tiered
preference pairs. The authors demonstrate that PLaID++ achieves state-of-the-art
stability and S.U.N. rates for both unconditional and space-group-conditioned generation,
while also proving that joint training across these tasks is mutually beneficial.

**Compliance With Llm Reviewing Policy:**

Affirmed.

**Final Justification:**

In the following discussion, the authors show the potential of the PLaID++ system and some preliminary results on band gap generation. As my concerns are addressed, I decide to raise my score to weak accept.

**Key Questions For Authors:**

### Q1. Internal Alignment (SFT+DPO) vs. External Agentic Paradigms.
Recent advances in LLM-driven materials discovery often leverage agentic frameworks (e.g., ReAct) combined with external tools (simulators or databases) for closed-loop iterative refinement. How do the authors compare the trade-offs between PLaID++'s pure weight-tuning approach (SFT + RLIP) and these external interactive agent paradigms? Furthermore, have the authors considered integrating both paradigms to further push the boundaries of complex structure generation, such as employing the RLIP-aligned model as the core policy network within an agentic framework?

### Q2. Potential Data Leakage and Memorization in Conditional Generation
In Section 4.2, the authors evaluate conditional generation on 7 space groups due to their sparsity in the training set (<1,000 samples). However, since these groups were still present during pre-training and SFT, concerns regarding data leakage and memorization arise. Can the authors clarify whether the model genuinely generalizes underlying symmetry rules, or merely retrieves and slightly modifies structural motifs seen in the training data? To rigorously prove true generalization over memorization, have the authors considered a strict hold-out evaluation (e.g., completely excluding specific space groups during fine-tuning to test zero-shot conditional generation)?

### Q3. Pipeline Extensibility to Complex Functional Properties
The title emphasizes "Targeted Design," yet the main text primarily demonstrates conditional generation on
geometric space groups. Real-world materials design typically targets complex physical or electronic properties (e.g., specific band gaps, Fermi levels, or thermal conductivity). What specific adjustments would the PLaID++ pipeline require to target such functional properties? Additionally, since RLIP currently relies heavily on computationally cheap MLIPs to construct preference pairs, how would the efficiency and feasibility of this alignment framework be maintained for complex properties that lack fast, low-cost
surrogate evaluators?

**Limitations:**

The authors do not explicitly discuss the limitations of their work in the manuscript. Please refer to the 'Weaknesses' and 'Questions' part for the limitations identified.

**Strengths And Weaknesses:**

**Strengths**
1. The alignment workflow is effective. The paper successfully adapts a proven NLP alignment pipeline (SFT + DPO) to the domain of materials science. This enables LLMs to reliably generate stable crystal structures under both unconditional and space-group-conditioned settings.
2. The experimental design is robust. The authors employ joint training on mixed datasets, dynamic temperature scaling to prevent mode collapse, and using distinct MLIPs to avoid reward hacking. The ablation studies convincingly validate these design choices.

**Weaknesses**
1. The baselines are insufficient. While the paper compares against the early CrystalLLM, it overlooks recent and highly relevant pure-LLM works (e.g., CrystalICL) and fails to benchmark against current state-of-the-art general LLMs (e.g., GPT-4o) in zero-shot or few-shot settings.
2. The claim of targeted design seems oversold. While the title promises "targeted" materials design, the scope of conditional generation remains relatively narrow. The authors successfully demonstrate targeting for 7 space groups and a high bulk modulus. However, to fully realize the promise of targeted design, further validation across more diverse practical scenarios, such as targeting electrical properties like band gap, is still needed.

---

> ### Author Rebuttal · Authors · 2026-03-31
>
> We thank the reviewer for their thoughtful review. We appreciate your recognition of the strong performance of our methods and our comprehensive ablations.
> > **The baselines are insufficient… (Missing CrystalICL, general LLMs)**
>
> Out-of-the-box LLMs struggle at crystal generation due to a lack of relevant training corpus data. We tried 1-shot prompting o3 to follow a prompt template similar to CrystalICL to generate 1000 crystals and obtained **a stability rate of 5% and a SUN rate <1%** as judged by eSEN.
>
> The recently released LeMat-Bench independently validates the core contributions of our work. LeMat-Bench compares 12 models, including Wyckoff-based methods similar to CrystalICL (e.g., WyFormer, CrystalFormer). **PLaID++ produces SUN crystals at >2x the next best model**, showing that our RLIP framework & joint training strategy are critical to performance.
>
> Regarding CrystalICL, they report only validity and coverage metrics which lack a proper evaluation of thermodynamic stability and novelty which are central to real-world feasibility. On these same validity proxy metrics, PLaID++ outperforms CrystalICL.
> |Method|Valid(Struct)|Valid(Comp)|Stab%|SUN%|
> |-|-|-|-|-|
> |CrystalICL(XYZ)|97.92|89.22|–|–|
> |CrystalICL(SGS)|98.59|86.55|–|–|
> |PLaID++|**99.88**|**97.34**|22.27|7.74|
>
> Also, CrystalICL's core contribution is its domain-informed few-shot ICL mechanism. This is orthogonal to our approach, and applying it to PLaID++ may further improve our results. We will include a discussion of CrystalICL in the revised paper.
> > **The claim of targeted design seems oversold…**
>
> The focus of our paper is to propose a novel framework for guiding generation towards desirable downstream properties. Our experimental scope matches the existing literature for conditional generation. Most other methods evaluate on only one or two target properties (CrystalFormer, DiffCSP++) and use different benchmarks that make cross-paper comparison difficult. To our knowledge, none besides MatterGen evaluates conditional generation with SUN. PLaID++ evaluates on three distinct conditioning types—categorical (space group), scalar (bulk modulus), and joint multi-objective (SUN + property constraint)—which span all ways one would use conditional generation.
>
> Moreover, our evaluation is more demanding than prior work: each conditionally generated crystal must satisfy both the target property and SUN. Without this joint requirement, conditional metrics alone cannot distinguish genuine targeted discovery from memorization or trivial perturbation. We open-source our full evaluation pipeline to encourage the community to adopt SUN-based evaluation for conditional generation going forward.
> > **Q3**
>
> RLIP is inherently an offline alignment procedure, so preference pairs can be created using expensive evaluators (DFT) without affecting training stability or runtime. As discussed in response to Reviewer 3, we empirically show RLIP works with any property you condition on and learns from _relative_ preferences, making it robust to both the evaluator and computational cost.
> > **Q1**
>
> Agent methods such as LLMatDesign and MatAgent require 10–32 sequential LLM API calls per crystal, each gated on surrogate evaluation feedback. These methods take minutes to generate each candidate. Meanwhile, PLaID++ generates 27.17 SUN crystals per minute (~2.2 seconds per SUN crystal). Hence, PLaID++'s pure weight-tuning approach trades per-sample interactive feedback for higher throughput. This throughput outweighs interactive feedback in real-world materials discovery: ML guided synthesis reports ~16% success rates (Cao et al., 2026) even after pre-screening computational targets, demonstrating the need for large candidate pools.
>
> Regarding the hybrid paradigm: **yes**, this is a promising direction for future work. PLaID++ could serve as the core policy within an agentic loop. However, this integration targets a different objective (iterative single-candidate refinement) than PLaID++'s contribution (high-throughput targeted generation with high SUN rates), and we consider it complementary rather than a limitation of the present submission.
> >**Q2**
>
> In Fig. 9 and 10 of the Appendix, we tested PLaID++’s space-group conditioned generation capabilities for 8 space groups excluded from the set of space groups we directly optimized for in our RLIP pipeline, and saw a 20% improvement in SSUN compared to fine-tuning alone. This demonstrates that RL is beneficial for tasks not explicitly incentivized by our reward structure, even for rare space groups in MP-20. For example, $P2_{13}$ with only 67 training occurrences saw a 26% improvement in SSUN. We did not hold out space groups, but PLaID++ generalizes to materials with larger unit cells than exist in training. 5.66% of the crystals generated by our PLaID++ model have > 20 atoms vs 0.08% for our 3D coord representation. This demonstrates how our Wyckoff representation helps the model to generalize from small unit cells.

---

> > ### Author Rebuttal · Reviewer_fnZy · 2026-04-04
> >
> > Thank you for the attempt to provide more experiments. However, I still have concerns about the scope of conditions, especially the claim for `empirically works with any conditions'. I maintain my score.

---

> > > ### Author Response · Authors · 2026-04-07
> > >
> > > We appreciate your engagement with our rebuttal and acknowledge that our phrasing was too strong. Our intended point was that RLIP is a general *procedure* that can, in principle, be applied whenever candidate materials can be ranked by a property evaluator, such as an MLIP or DFT. We cannot claim our approach would successfully learn to generate any materials with any arbitrary property. However, we reiterate that our evaluation does cover a range of practical condition types, including: categorical (space group), scalar (bulk modulus), and joint multi-objective generation (S.U.N. plus a property constraint). We believe that the breadth of our evaluation is in line with other work in this space (CrystalFormer, DiffCSP++, Con-CDVAE).
> > >
> > > Nevertheless, to provide additional evidence that our pipeline is a general procedure, we ran a preliminary experiment extending PLaID++ to targeted band gap generation, an electronic property that requires a dedicated property-prediction GNN (MEGNet). We construct tiered preference pairs analogous to our bulk modulus pipeline, ranking crystals by high band gap ( $\geq$ 3.0eV), moderate band gap ($\geq$ 1.0eV, $<$ 3.0eV), and low band gap, in combination with stability and novelty. Even after a single RLIP iteration, PLaID++ shows clear improvement over the non-DPO baseline across 1000 generated samples:
> > >
> > > | Model | # Band Gap > 1 eV & SUN | # Band Gap > 2 eV & SUN | # Band Gap > 3 eV & SUN |
> > > |--------------------|----------|----------|----------|
> > > | Wyckoff Base (Non-DPO) | 4 | 1 | 0 |
> > > | PLaID++ (It 1) | 15 | 9 | 3 |
> > >
> > > These results reinforce that our RLIP pipeline can be successfully applied across a range of crystal properties. We were unable to run the full PLaID++ pipeline for band gap due to time and compute constraints, but we will include a multi-iteration ablation in the camera-ready.

---

### Official Review · Reviewer_rajw · 2026-03-13

**Soundness:** 3
**Presentation:** 3
**Significance:** 3
**Originality:** 3
**Overall Recommendation:** 5
**Confidence:** 2

**Summary:**

The paper introduces PLaID++, an alignment strategy for LLMs designed to generate stable inorganic crystal structures by casting materials design as a preference alignment problem. The authors use a Wyckoff-based text representation to allow for physical constraints like symmetry to be more capably handled by LLMs, thereby improving their ability to handle 3D geometry. DPO is used to guide the model towards thermodynamically stable structures according to physics-based rewards. The experiments demonstrate that the proposed approach generates valid materials at a significantly higher rate than competing approaches while allowing for the targeted design of crystals with specific symmetry properties and with greater computational efficiency.

**Compliance With Llm Reviewing Policy:**

Affirmed.

**Final Justification:**

The authors' response clarified some of what I missed about baseline comparisons addressed my concerns regarding novelty. This paper is a solid machine learning application to a relevant problem domain (materials science) and achieves competitive results against highly specialized baselines, which is a noteworthy accomplishment in its own right.

**Key Questions For Authors:**

- How well does modeling Wyckoff-based text representations scale when generating much larger unit cells or low-symmetry structures where the string length and complexity are increased?
- In going from what I imagine are XYZ coordinates to the Wyckoff-based text representations, how robust are the resulting LLM-extracted embeddings to "coordinate quantization" errors? I.e., does the discrete nature of text tokens limit the precision of the generated atomic positions compared to models that operate on the coordinates directly?

**Limitations:**

Yes

**Strengths And Weaknesses:**

Strengths:
- Well written, well motivated paper. Key results are presented clearly in nicely formatted figures and tables.
- Bridges material science with LLMs by casting crystal stability as a preference alignment problem.
- The authors report measureable improvement in the speed of generating stable, unique, and novel materials compared to earlier benchmarks, demonstrating that LLM-based architectures combined with symmetry-respecting representations can compete with specialized geometric models.

Weaknesses:
- Comparisons are largely limited to other LLM-based approaches published in the last few years. Without having much familiarity with the literature on materials discovery, I wonder if there are more traditional baselines that are worth comparing against.
- Novelty is mostly in the form of clever combination of existing components, e.g., prompting using Wyckhoff-based text representations, applying DPO with interatomic potentials.

---

> ### Author Rebuttal · Authors · 2026-03-31
>
> We thank Reviewer 1 for recognizing our cross-domain contribution, the computational efficiency of our model, and the ability of our general framework to outperform specialized models. Below, we address your main questions and concerns.
>
> >_**Comparisons are largely limited to other LLM-based approaches published in the last few years.**_
>
> We clarify that our comparisons are not limited to LLM-based approaches. Specifically, Table 1 benchmarks PLaID++ against three distinct model families: variational autoencoders (CDVAE [ICLR 22]), diffusion and flow-based models (FlowMM [ICML 24], MatterGen [Nature 25]), and other language models (FlowLLM [NeurIPS 24], ADiT [ICML 25]). PLaID++ outperforms all of these across stability and S.U.N. metrics.
>
> Regarding more traditional baselines, as noted in our conclusion, ab initio random structure searching (AIRSS) (Pickard & Needs, 2011) achieves a <1% success rate in identifying stable materials—far below PLaID++'s 22.27%.
>
> >_**Novelty is mostly in the form of clever combination of existing components…**_
>
> Although the domains in which post-training has been applied are rapidly expanding, it is not obvious that RL would generalize well for generating *novel* inorganic crystal structures. This is because generating novel structures is inherently an out-of-distribution task, which RL generally struggles with.
>
> Importantly, we show that a structured Wyckoff representation is not merely a different crystal formatting. Rather, we see that the coordinate-only representation’s S.U.N. rate collapses across DPO iterations despite higher stability rates; it is only with innovation at the representation level that we see generalization rather than memorization under preference optimization.
>
> Additionally, our aim is not architectural novelty for its own sake; instead, we focus on the components that actually amplify high-throughput crystal discovery. Our results directly speak to the bitter lesson, the idea that data scaling regimes outperform hand-crafted, bespoke models. In crystal discovery, geometry-aware or equivariant architectures (e.g., Wyckoff-Transformer-style models that embed Wyckoff sites as learned tokens) exist, yet PLaID++ outperforms them on all metrics relevant to discovery. This shows that algorithmic novelty is not the primary bottleneck for materials discovery; the core challenge is how to represent crystals and align models with physics-derived objectives.
>
>
>
> >_**How well does modeling Wyckoff-based text representations scale … ?**_
>
> We use a Wyckoff representation since it scales better than coordinate-based alternatives: atoms sharing a Wyckoff site only need to increment one digit rather than new coordinate lines, so token count grows with the number of unique sites rather than total atoms. To illustrate this, we tokenized MP-20 crystals at varying atom counts and symmetries using PLaID++’s Qwen tokenizer. We will add this analysis to the camera-ready paper:
>
> |Symmetry|Avg 3D Coord Tokens|Avg Wyckoff Tokens|Avg Reduction|
> |-|-|-|-|
> |Low (Monoclinic)|211|174|17%|
> |Moderate (Orthorhombic/Tetragonal)|230|147|36%|
> |High (Cubic/Hexagonal)|214|145|32%|
> |**Overall**|**218**|**155**|**29%**|
>
> Wyckoff encodings save tokens across all symmetries, with reductions exceeding 32% at higher symmetries. Even for low-symmetry structures (e.g., P1), the Wyckoff encoding converges to the coordinate baseline. Since the vast majority of known inorganic materials exhibit nontrivial symmetry, the representation remains advantageous in practice.
> >_**How robust are the resulting LLM-extracted embeddings to "coordinate quantization" errors?**_
>
> Models that use custom Wyckoff embeddings or operate on continuous coordinates, such as WyckoffTransformer (Kazeev et al., 2024) and CrystFormer (Cao et al., 2024), mitigate “coordinate quantization” errors. However, despite this architectural difference, PLaID++ outperforms both on stability and S.U.N. rate. This aligns with the broader "bitter lesson" observation: leveraging the pretrained knowledge and scale of a general-purpose LLM with a simple text tokenizer outperforms bespoke architectures with custom embeddings, even when those architectures have theoretically higher coordinate precision.
>
> Moreover, in any practical crystal discovery pipeline, generated structures are always relaxed post-generation using MLIPs or DFT to find the nearest energy minimum. This relaxation step corrects small coordinate errors. The goal of the generative model is therefore not to produce exact atomic positions, but to produce a diverse set of candidates that reach stable minima after relaxation. Our Wyckoff coordinates provide sufficient resolution for this purpose, as validated by our DFT results (Section 5.3), where 19.1% of relaxed structures are confirmed stable.

---

> > ### Author Rebuttal · Reviewer_rajw · 2026-04-02
> >
> > I thank the authors for their response to my questions and concerns. I am increasing my score from 4 to 5.

---

### Decision · Program_Chairs · 2026-04-30

**Decision:**

Accept (regular)

**Comment:**

This is a well-motivated paper using preference optimization-based fine-tuning to improve LLM performance at generating inorganic materials, relying on clever use of a Wyckoff representation. The reviews emphasize that, while the approach lacks substantive technical novelty, the results are empirically strong, and the paper is clear and well presented. The paper is nicely situated in the broader discussion of performance of discrete, tokenized outputs from general-purpose LLMs in comparison to tailored architectures.